# Applications of Serious Games as Affective Disorder Therapies in Autistic and Neurotypical Individuals: A Literature Review

**Fahad Ahmed** [1,2,*] , **Jesús Requena Carrión** [1] , **Francesco Bellotti** [2,*] , **Giacinto Barresi** [3] , **Federica Floris** [4] **and Riccardo Berta** [2]

1 School of Electronic Engineering and Computer Science, Queen Mary University of London, Mile End Rd., Bethnal Green, London E1 4NS, UK
2 Department of Electrical, Electronic and Telecommunication Engineering (DITEN), University of Genoa, Via All'Opera Pia, 15, 16145 Genoa, Italy
3 Rehab Technologies Lab., Istituto Italiano di Tecnologia, Via Morego, 30, 16163 Genoa, Italy
4 Piccolo Cottolengo Genovese di Don Orione, Via Benvenuto Cellini, 22, 16143 Genova, Italy
* Correspondence: fahadahmed@qmul.ac.uk (F.A.); francesco.bellotti@unige.it (F.B.)

**Abstract:** Affective disorders can greatly influence the everyday lives of neurotypical and autistic individuals. As platforms that promote engagement, computer-based serious games (CSGs) have been previously proposed as therapies to treat affective disorders for both populations. However, these CSGs were assessed on a wide variety of experimental conditions, and there is a lack of comparative studies on their effectiveness. In this study, we identified and analyzed 37 interventions of CSGs for affective disorders in autistic and neurotypical individuals from 507 initial search results from four databases (Embase, Scopus, Web Of Science and IEEE Xplore), using concepts such as 'serious-games', 'affective-disorders', 'autism' and 'neurotypical'. A total of 21 different CSGs were identified that were evaluated with 30 different outcome measures in the reviewed interventions. A positive impact was reported in 22 of them; specifically, all instances of depression interventions reported a positive impact of CSG therapies. Our comparative analysis indicates that CSG applications could be effective in treating affective disorders in autistic and neurotypical individuals. Additionally, our analysis identifies CSG design characteristics that might be useful in applications involving depression, anxiety and phobias. Based on these characteristics, we provide a set of recommendations for CSG interventions for affective disorder therapies.

**Keywords:** affective computing; serious games; autism; neurotypical; affective disorder

## 1. Introduction

Human emotions play an integral part in nearly all the decisions we make. They have far-reaching impacts on our daily activities, ranging from how we behave, what we wear, and what we eat to which investments should we make [1–4]. In other words, both incidental and integral emotions play a vital role in setting our default preferences in everyday decision-making tasks [5–7]. An incidental emotional state is often termed as 'mood', reflecting the general state of our minds and is not dependent on any judgments or decisions taken by our conscious selves. On the other hand, integral emotions are the ones that arise due to the conscious evaluation of the outcomes of our decisions [8]. Studies have shown that the emotional health of an individual has direct correlations to one's mental health [9,10], which in turn means that emotion regulation can be a viable means of maintaining not only the mental health of individuals [11,12] but also curbing adverse physiological manifestations such as prolonged exposure to negative emotional states [13].

Affective disorders have been shown to produce telltale signs of abnormal incidental as well as integral emotional response and decision-making patterns that occur in individuals having different neurological make-ups, for instance from neurotypicals [14–17] to people having autistic traits [18,19]. Individuals with autistic traits are considered to have autism spectrum disorder (ASD). Autism spectrum disorder (ASD) is a neurological

and developmental disorder that affects how people interact with others, communicate, learn, and behave. Although autism can be diagnosed at any age, it is described as a "developmental disorder" because symptoms generally appear in the first two years of life. According to the Diagnostic and Statistical Manual of Mental Disorders (DSM-5—2022) [20], a guide created by the American Psychiatric Association that health care providers use to diagnose mental disorders, people with ASD often have the following: (1) difficulty in communication and interaction with other people, (2) restricted interests and repetitive behaviors, and (3) symptoms that affect their ability to function in school, work, and other areas of life. These functioning characteristics determine adaptation difficulties in people with ASD, difficulties that significantly affect their mental well-being. The general trend observed between both autistic and neurotypical individuals with affective disorders include emotional inflexibility [18] and deficit in emotion regulation [14,17,18,21–23]. Numerous pieces of research have found that people with ASD are more likely to develop a psychiatric disorder, such as affective disorders, than the general population [24]. Among the different types of affective disorders, the mortality rate is relatively higher in depression, closely followed by bipolar disorder and then by anxiety disorder [25]. Depressed autistic individuals are also three times more likely to commit suicide than the general population of the same age and sex demographic [26] and a later study in 2014 estimated that around 50% of autistic individuals suffer from one form of major depressive disorder [27,28].

A number of studies have shown that individuals with ASD, especially younger ones, show greater engagement with serious video games on interactive devices such as smart phones and tablet computers when compared to IQ-matched neurotypical people of the same age and gender demographics [29–32]. Furthermore, there are a lot of studies that investigate the applications of certain computer-based serious games (CSGs) for certain affective disorders for both neurotypical and autistic populations [33–68]. However, with such varieties of studies in the different types of applications of CSGs for different affective disorders in diverse population demographics, it is vital for an objective, comparative study of these applications to understand the merits of serious applications for autistic and neurotypical individuals with affective disorders. Additionally, most of the reviews of CSG applications often divulge towards applications of CSGs for academic performance enhancements [69–72], for specific disorders such as ADHD [73], only depression [74,75], only anxiety [76], were scoping reviews and hence had taken a shallow approach to investigating the applications [77,78] or investigated for mental disorders other than affective disorders [79–83] and none of these reviews investigated the comparative effectiveness of specific game design characteristics for different affective disorders. Since comparative studies are an effective means of analyzing characteristic differences between studies, objects or subjects, they are useful for identifying the effectiveness of applications in a given field [84]. Comparing studies with such diverse populations as well as heterogeneous data collection and experimental techniques presents its own set challenges and could usher in a clear understanding of whether CSGs can be effectively applied as therapeutics for autistic and neurotypical individuals with affective disorders. To our knowledge, no such analysis, either quantitative or qualitative, exists that investigates the current state-of-the-art applications of CSGs for therapeutical purposes for autistic and neurotypical individuals with affective disorders; therefore, there is a lack of comparative studies of this nature in the literature. Hence, in this paper, we present a literature review in which we investigated the current state-of-the-art of applications of CSGs for affective disorder therapeutics for autistic and neurotypical individuals.

To understand the effectiveness of CSGs as therapies for affective disorders, it is important to understand which individuals with affective disorders have had a positive impact from a CSG intervention [85]. Furthermore, it is important to know the design characteristics of the CSGs that were actually effective in their applications for each affective disorder in the articles to be reviewed for this study. This will enable us to identify the appropriate game design characteristics for different affective disorders. Hence, we devised the following research questions for this review:

1.  Can CSGs be effective as therapies for affective disorders in autistic and neurotypical individuals?
2.  For which affective disorders among autistic or neurotypical individuals have CSGs been effectively applied?
3.  What are the design characteristics of the games that are effective for different affective disorders?

The structure of the paper is as follows. Section 2 discusses the methodology of our study that was used to obtain and analyze the reviewed articles. Section 3 explains the analysis that was performed on the reviewed articles. The results of this analysis are presented in Section 4. Finally, Section 5 wraps up our study and provides recommendations on viable game design characteristics for CSGs to be used as therapies for different types of affective disorders.

## 2. Review Methodology

The protocol for this review was developed in six stages; the first three were searching stages and the next three were screening phases. Figure 1 shows the stages of the protocol we followed to obtain the articles to be reviewed.

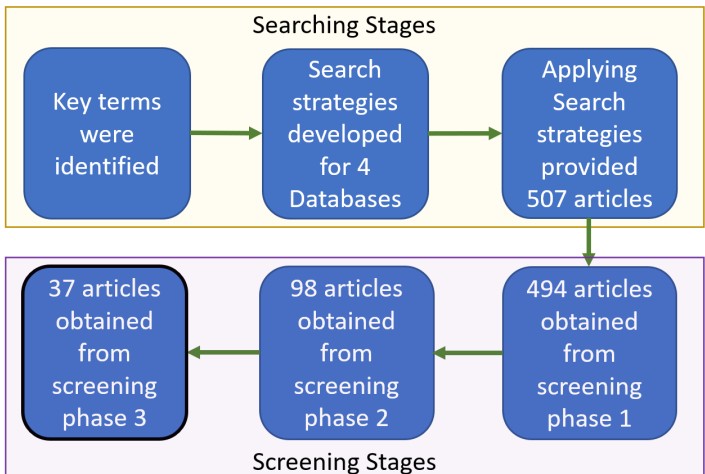

**Figure 1.** Six stage protocol implemented to obtain the articles for this literature review.

In the first stage, the key terms were identified that encompassed the basic concepts we wanted to investigate, such as 'affective disorder', 'serious games', 'autism' and 'neurotypical', as well as their synonyms from the Emtree hierarchy [86], such as 'mood disorder', 'affective illness', 'videogame' and 'VR simulator'. Then, in the second stage, the queries implementing our search strategies were developed using these key terms and their synonyms, as shown in Table 1. Afterward, in the third stage, the search strategies were applied to four scientific databases (IEEE [87], Embase [88], Web Of Science [89] and Scopus [90]); according to the guidelines provided by AMSTAR [91], searching in at least two databases should be sufficient for reviewing the literature in a field. At this stage, we obtained 507 articles from the database searches. The fourth through sixth stages correspond to the three phases in which these articles were screened. At the fourth stage, phase one of the screening commenced where the duplicate articles, or articles that appeared in more than one of the four databases we searched in, were eliminated, which left us with 494 articles. Then, in stage five, phase two of screening was undertaken and here, articles were screened based on their titles according to our eligibility criteria, which are further discussed in Section 2.1. A total of 98 articles were obtained, which were then further screened in the same manner based on their abstracts in stage six, which implemented screening phase three. After this stage, the final 37 articles that were reviewed for this study were obtained. Stages four through six are discussed in further detail in Section 2.2. Section 2.3 discusses

the key points of analysis that we focused on to perform the comparative analysis of the extracted data about the reviewed studies.

**Table 1.** Search strategies for the four databases.

| Database | Search Strategy | Initial Articles | Title Screened Articles | Abstract Screened Articles |
|---|---|---|---|---|
| Embase | ('mood disorder'/exp/mj OR 'affective disorder':ti,ab,kw OR 'affective disorders':ti,ab,kw OR 'affective disturbance':ti,ab,kw OR 'affective illness':ti,ab,kw OR 'mood disorder':ti,ab,kw OR 'mood disorders':ti,ab,kw OR 'mood disturbance':ti,ab,kw OR 'mood disturbances':ti,ab,kw OR 'major depression'/exp/mj OR 'depression, major' OR 'depression, unipolar' OR 'depressive disorder, major' OR 'major depression' OR 'major depressive disorder' OR 'major depressive episode' OR 'unipolar depression' OR 'unipolar disorder' OR 'bipolar disorder'/exp/mj OR 'bipolar affective disorder' OR 'bipolar and related disorders' OR 'bipolar disorder' OR 'bipolar illness' OR 'bipolar psychosis' OR 'depression, manic' OR 'manic depression' OR 'manic depression psychosis' OR 'manic depressive' OR 'manic depressive disease' OR 'manic depressive disorder' OR 'manic depressive illness' OR 'manic depressive psychosis' OR 'manic depressive reaction' OR 'manic depressive syndrome' OR 'maniodepressive psychosis' OR 'mano depressive syndrome' OR 'psychosis, manic depressive' OR 'anxiety disorder'/exp/mj OR 'anxiety disorder' OR 'anxiety disorders') AND ('serious games':ti,ab OR 'video game'/exp/mj OR 'tv games' OR 'computer game' OR 'computergame' OR 'television game' OR 'video game' OR 'video games' OR 'videogame' OR 'videogames' OR 'virtual reality simulator'/exp/mj OR 'vr simulation system' OR 'vr simulator' OR 'virtual reality simulation devices' OR 'virtual reality simulation system' OR 'virtual reality simulator') AND ('article'/it OR 'conference paper'/it)AND [2008–2023]/py | 255 | 60 | 30 |
| Scopus | ( TITLE-ABS ((ÄFFECTIVE DISORDER" OR "MOOD DISORDER" OR ÄNXIETY" OR "DEPRESSION") AND (SSERIOUS GAMES" OR "VIDEO GAMES" OR "COMPUTER GAMES" OR "VR" OR "VIRTUAL REALITY" OR ÄR") AND (ÄUTISM" OR ÄUTISTIC" OR "NEUROTYPICAL"))) AND (LIMIT-TO ( DOCTYPE, är") OR LIMIT-TO (DOCTYPE, "cp") OR LIMIT-TO (DOCTYPE, "bk") OR LIMIT-TO (DOCTYPE, "ch")) AND (LIMIT-TO (PUBYEAR, 2023) AND (LIMIT-TO (PUBYEAR, 2022) OR LIMIT-TO ( PUBYEAR, 2021) OR LIMIT-TO (PUBYEAR, 2020) OR LIMIT-TO (PUBYEAR, 2019) OR LIMIT-TO (PUBYEAR, 2018) OR LIMIT-TO (PUBYEAR, 2017) OR LIMIT-TO (PUBYEAR, 2016) OR LIMIT-TO (PUBYEAR, 2015) OR LIMIT-TO (PUBYEAR, 2014) OR LIMIT-TO (PUBYEAR, 2013) OR LIMIT-TO (PUBYEAR, 2012) OR LIMIT-TO (PUBYEAR, 2008)) | 71 | 16 | 5 |

**Table 1.** *Cont.*

| Database | Search Strategy | Initial Articles | Title Screened Articles | Abstract Screened Articles |
|---|---|---|---|---|
| Web of Science | (((TI = (SSerious Games") OR TI = (Video Games) OR TI = (Computer Games) OR TI = (VR) OR TI = (Virtual Reality Games)) OR (AB = (Serious Games) OR AB = (Video Games) OR AB = (Computer Games) OR AB = (VR) OR AB = (Virtual Reality Games)))) AND ((TI = (Affective disorder) OR TI = (mood disorder) OR TI = (depression) OR TI = (bipolar) OR TI = (anxiety)) OR (AB = (depression) OR AB = (Affective disorder) OR AB = (mood disorder) OR AB = (bipolar) OR AB = (anxiety))) AND ((TI = (AUTISM) OR TI = (AUTISTIC) OR TI = (Neurotypical)) OR (AB = (AUTISM) OR AB = (AUTISTIC) OR AB = (Neurotypical)) ) | 48 | 11 | 2 |
| IEEE Xplore | (( NOT (("Full Text Only":REVIEW) OR ("Full Text Only": SURVEY) OR ("Full Text Only":REPORT)) AND (((("Full Text Only":SERIOUS GAME) OR ("Full Text Only":VIDEO GAME) OR ("Full Text Only": VR) OR ("Full Text Only": VIRTUAL REALITY) OR ("Full Text Only": MOBILE GAME)) AND (("Full Text Only":AFFECTIVE DISORDER) OR ("Full Text Only": MOOD DISORDER) OR ("Full Text Only": BIPOLAR DISORDER) OR ("Full Text Only": DEPRESSION) OR ("Full Text Only": ANXIETY)) AND (("Full Text Only": AUTISM) OR ("Full Text Only": AUTISTIC) OR ("Full Text Only": NEUROTYPICAL)))))) | 133 | 12 | 0 |

### 2.1. Eligibility Criteria

According to the Cochrane Handbook for Systematic Reviews of Interventions (version 6.3), the eligibility criteria must include the inclusion and exclusion criteria that are true to the scope and objectives of a review [92]. In our study, for inclusion the articles needed to:

- Have investigated digital game-based intervention or the usage of games for detection, diagnosis, prognosis, therapeutics of affective disorders or its effects on individuals having said disorders
- Be a published journal, conference article, book, or book chapter;
- Be written in English;
- Be published in the last 15 years;
- Have participants.

Furthermore, for exclusion, the articles needed to:

- Be reviews, surveys or reports;
- Not use computer-based (either VR, portable or desktop) games;
- Only address aspects of affective disorders indirectly, for instance, where affective disorders were not the main ailment for the participants and were only treated as a consequence of some other illness.

### 2.2. Article Screening and Data Extraction Process

All the search strategies were applied to the databases on 4 January 2023 to obtain a list of articles to be reviewed. The publication year restrictions were provided in the

search strategies for Scopus and Embase while for Web Of Science and IEEE Xplore, due to the limitations of their search query syntax, the search results were filtered using the date filter on the databases' websites from their respective search result pages. A total of 507 articles (Embase: 255, Scopus: 71, Web Of Science: 48, IEEE Xplore: 133) were obtained from the search results from these 4 databases. The search results from the databases were exported into CSV format files keeping the metadata selection as similar as possible across the databases, yet there still were some differences between the structuring of the information of the articles in the exported search results from different databases. For instance, the export files for some databases contained empty columns for the metadata headers which had not been selected during the exporting of the search results. To ensure the consistency of the metadata across all search results, the exported search results were merged together into one Microsoft (MS) Excel file (software version 2211) that contained all of the metadata for the 507 search results. Afterward, these search results were screened in three phases based on the eligibility criteria described in Section 2.1.

In the first screening phase, the duplicates were manually deleted from the MS Excel file using the 'conditional formatting' functionality of the software to detect the duplicates. Then, in the second screening phase, the titles were examined to determine their relevance to our research questions using our eligibility criteria and at this stage, all the articles from the search results that were articles about non-digital game-based interventions, were surveys, or were about managing comorbidities of affective disorders were excluded. After the second screening phase, we obtained a shortlist of 98 articles (Embase: 59, Scopus: 16, Web Of Science: 11, IEEE Xplore: 12). Afterwards, in the last phase, the selected articles were screened based on the relevance of their abstracts, again using our eligibility criteria and at this phase all articles that were surveys, reviews and/or reports but did not mention that in the title were screened out. Furthermore, some other articles, which seemed to align with our goals from their titles in the second phase but no longer seemed relevant based on their abstracts, were removed from our final list of articles to be reviewed. The third screening phase produced 37 articles (Embase: 30, Scopus: 5, Web Of Science: 2, IEEE Xplore: 0). These were the final selection of articles being reviewed for this study.

In order to extract the information that was relevant to our research questions, a custom table was prepared in MS Excel with 18 columns and the relevant information from the selected articles was inserted into this table. The columns in this table represented three key aspects of the studies, namely information about the game used in the study, information about the participants in the studies and the different research characteristics of the studies. The information from this table was separated into three tables and is presented in Table 2 (game information), Table 3 (participant information) and Table 4 (research characteristics).

*2.3. Key Points of Analysis*

Since the population demographics were very diverse (further discussed in Section 3.1), the studies that had at least one similarity in their evaluation techniques for the same type or comorbidity of affective disorder were deemed to be comparable. Hence, we compared the game design characteristics of the game they used and the research characteristics (experimental methodologies, data collection and evaluation techniques, as well as outcomes of the studies). The key points of analysis were as follows:

1.  Game design characteristics:
    The hallmark of an effective CSG is for it to align with the three pillars of the triadic game design methodology [93], which are 'reality' (referring to what real-world issue the CSG solves), 'meaning' (referring to the purpose of the CSG as well as the context of the game) and 'play' (referring to how the interaction happens between the players and the game). The 'reality' for a CSG can be obtained from the application of the game in a study, e.g., if it is used for reducing the symptoms of depression in neurotypical adolescents [93]. The 'meaning' of a game can be partially obtained from the application of the game in a study, e.g., the purpose of a game could be a therapeutic

experience to individuals with anxiety and autism, and partially through the target populations' profile as well as gameplay type (single player vs. multiplayer) [93]. Finally, the 'play' of a game can be obtained from the game design characteristics (genre, player perspective, game dimensionality and mechanics) as all of these have a direct impact on how the player interacts with the game [93]. The game genre classification was based on the revised genre map developed for games [94], with some added genres that are discussed further in Section 3.1. Therefore, we compared the game design characteristics of the different games that were used in the selected studies.

2. Experimental methodologies:
   As stated in [95], "A proper experimental design serves as a road map to the study methods, helping readers to understand more clearly how the data were obtained and, therefore, assisting them in properly analyzing the results". Hence, it will be important for us to know about the experimental methodology used in the studies in order to understand the significance of the findings and to have them compared.

3. Data collection and evaluation techniques:
   Data collection and evaluation techniques are paramount to understanding the research methodology of a study [96,97]. Furthermore, studies that used the same evaluation techniques, even amongst an ensemble of other techniques, could be comparable in terms of their findings.

4. Outcomes of the studies:
   It is necessary to comparatively analyze the effectiveness of a particular application of a CSG for a specific affective disorder and for a specific population [80]. As, based on the outcomes of the selected studies, it could be determined whether the CSG application would be effective or not, comparing the outcomes of the selected studies is necessary to satisfy the research questions of this study.

**Table 2.** Details of the Serious Games used as interventions in the selected studies.

| Game Name | Platform | Game Genre | Player Perspective | Game Dimensionality | Description of Gameplay | Game Mechanics | Control Group's Game | Citation |
|---|---|---|---|---|---|---|---|---|
| Neuro Racer | Ipad OS | Racing | First Person | 3D | The game involves guiding a character through a continuous road while responding to select targets. | ->Controlling a car on a road ->Avoiding obstacles as they appear | Connecting letters in a grid on the screen horizontally, vertically, or diagonally to form as many words of at least two letters as possible during a 25-min period [98], NA for [37,56] | [37,56,98] |
| Not given | Roblox | Music | Isometric | 2D | Players had to press certain areas on the screen based on rhythm being played and on-screen cues. | ->Hitting targets ->Targets appear based on the music being played | NA | [33] |
| Pegasas VR | Not given | Social VR | Immersive | 3D (VR) | The VR game had AI virtual agents that conversed with the player in their school's setting. | ->No real reward mechanism ->Player exposed to socialisation scenarios | NA | [34] |
| MindLight | Microsoft Xbox 360 [35,55], Windows [49], Not Given for [62,64] | Adventure | Third Person | 3D | The game centers on Arty, who must navigate through his grandmother's scary mansion with the help of a glowing hat named Teru. | ->The light is essential for navigation and survival of the player ->Brightness of light depends on player's relaxation levels ->Has several antagonists the player must survive from | NA [35], Same, just that the 2nd phase trail is considered the control group [49], NA for [35,49,55,62], Tiple | [35,49,55,62,64] |

**Table 2.** *Cont.*

| Game Name | Platform | Game Genre | Player Perspective | Game Dimensionality | Description of Gameplay | Game Mechanics | Control Group's Game | Citation |
|---|---|---|---|---|---|---|---|---|
| ACT RAGE-Control | Not given | Action | Isometric | 2D | Players will fly a spaceship and must control their heart rate, measured by pulse oximeter, to allow their spaceship to fire. If a player's heart rate exceeds baseline by 7 bpm, their spaceship will fire blanks. These blanks do not destroy the asteroids and are accompanied by a different sound. | ->Player heartrate impacted the win probability ->Ship control was voluntary-Had reward score | Identical to ACT RAGE-Control but the heart rate does not effect the gameplay. | [36] |
| Not given | VR TierOne | Therapeutic VR | Immersive | 3D (VR) | Virtual Therapeutic Garden where the participant is able to calm down and relax. | ->Move around in virtual world ->Player could change environment | NA | [38] |
| Vrelax | Android, Samsung Gear VR | Therapeutic VR | Immersive | 3D (VR) | Participants could navigate between interconnected 360-degree video nature environments such as beaches, underwater coral reefs, mountain meadows, etc. by looking at hotspots that were activated after 2 s. | ->Move around in virtual world ->Player could change environment by going to hotspots | NA | [39] |
| Tetris | Nintendo DS XL [45], Not Given [40] | Puzzle | Isometric | 2D | Arranging the orientation of different shaped blocks that keep falling from the top of screen, in order to fit them without gaps to eliminate them. | ->Player can change orientation of falling objects ->After leveling off the latest layer, the player gets more 'life' | NA | [40,45] |

**Table 2.** *Cont.*

| Game Name | Platform | Game Genre | Player Perspective | Game Dimensionality | Description of Gameplay | Game Mechanics | Control Group's Game | Citation |
|---|---|---|---|---|---|---|---|---|
| Bear Blast | Oculus VR | Action-Shooter | Immersive | 3D (VR) | Participant throughs objects to pop virtual bears. | ->Can shoot targets with different objects<br>->Had reward scores | Candy crush | [41] |
| BIPOLIFE | Web | Social Simulation | Isometric | 3D | The participant controls an avatar who has BD and acts in a variety of everyday life situations. | ->Player can interact with NPCs through dialogues<br>->Game has alternate storylines based on player decisions<br>->No real reward mechanism | NA | [42] |
| Not given | iOS | Social VR | Immersive | 3D (VR) | Participant interacted with virtual classmates in a classroom setting. | ->Player exposed to socialisation scenarios | NA | [43] |
| Plants vs Zombie | Not Given | Action-Shooter | Isometric | 3D | Mutated plants are used to shoot down Zombies. | ->Typical shooter mechanic<br>->Player can shoot zombies with plants | NA | [44] |
| VIMSE | Android, Samsung Gear VR | Therapeutic VR | Immersive | 3D (VR) | The participant was exposed to increasingly realistic spiders, just as in OSTs. | ->Player could interact with virtual spiders<br>->Spiders become and more realistic with each level | NA | [46] |
| Not given | Windows, Oculus Dev kit | Therapeutic VR | Immersive | 3D (VR) | Patients were exposed to five different researcher-controlled VR dental scenarios. | ->The patients were exposed to five different VR dental scenarios that were controlled by researchers using two networked computers and a head-mounted device. Patients had limited interaction ability within the virtual environment | NA | [47] |
| Space Academy | Windows | Therapeutic VR | Immersive | 3D (VR) | The application had several prebuilt virtual environments that the participant could explore. | ->Player could move around a virtual world<br>->Could change the environment as well | NA | [48] |
| Dance Central | Xbox 360 | Exergame | Immersive | 3D | Copying movements of on-screen characters that are tracked by Kinect devices. | ->Tracked player movement is compared with NPC movement to give reward score. | NA | [50] |

**Table 2.** *Cont.*

| Game Name | Platform | Game Genre | Player Perspective | Game Dimensionality | Description of Gameplay | Game Mechanics | Control Group's Game | Citation |
|---|---|---|---|---|---|---|---|---|
| Hit the Cancer | Android | Shooter | First Person | 2D | Different cells, either cancerous or not, keep on coming it the screen and the player has to shoot only the cancerous cells down. | ->Typical shooter mechanics ->Had reward points | NA | [51] |
| Pesky Gnats: The Feel Good Island | Android/iOS/ Windows | Adventure | Third Person | 3D | Explore the virtual island and interact with the locals. | ->Player can customise playing character ->Has scoring mechanism ->Can interact with NPCs through dialogues | Same, just that the 2nd phase trail is considered the control group | [52] |
| Claustrophobia Game | Windows, Oculus Rift | Exploration | First Person | 3D | The player can explore a 3D world and navigation pointers lead the player to confined spaces. | ->Move around in virtual world ->Following navigation cues leads to confined spaces | Same, just control group does not have claustrophobia. | [53] |
| 0Phobia | Android/iOS | Exploration and Action | Immersive | 3D (VR) | The player is exposed to several scenarios that his/her acrophobia in the game. | ->Move around in an interactive theatre ->Complete tasks related to acrophobia | Same, just that the 2nd phase trail with a different is considered the control group. | [54] |
| Not given | Windows, V8 HMD | Therapeutic VR | Immersive | 3D (VR) | The participant has several virtual places to move around in the virtual world. | ->No real reward mechanism ->Player exposed to socialisation scenarios | NA | [57] |
| Flowy | iOS | Adventure | Isometric | 2D | The application had several minigames with different dynamics. | Not Given | Same, just that the 2nd phase trail is considered the control group. | [58] |

**Table 2.** *Cont.*

| Game Name | Platform | Game Genre | Player Perspective | Game Dimensionality | Description of Gameplay | Game Mechanics | Control Group's Game | Citation |
|---|---|---|---|---|---|---|---|---|
| Maya | Web | Role-playing Adventure | First Person | 3D | The player controls the namesake protagonist who interacts with out spoken characters and takes part in socially challenging activities. | ->Player can interact with NPCs through dialogues<br>->Game has alternate storylines based on player decisions | NA | [59] |
| Processing Speed Training Game | Android | Puzzle | Isometric | 2D | A collection of games that include localization, detection, or identification elements. | ->Moving onscreen objects to solve puzzles | knowledge quiz training game. | [60] |
| Wii Sports gaming package | Nintendo's Wii Sports | Exergame | Immersive | 3D | Contained an assortment of sports oriented games. | ->Moving onscreen objects with Wii controllers to play sports games | NA | [61] |
| New Horizon and SpaceControl | Android/iOS | Exploration Puzzle | First Person | 2D | The protagonists travels through space and explores different planets. There were some additional mini games. | ->Ensamble organisation of games<br>-> Main game involves moving a ship in space with obstacles | NA | [63,67] |
| Not given | Merlin i-theatre | Not Given | Immersive | 3D (VR) | Not Given | Not Given | NA | [65] |
| Not given | Customised | SocialVR | Immersive | 3D (VR) | A blue room was used to project a virtual environment all around the participant and different scenes of different social situations were played with virtual interactring agents. | ->No real reward mechanism<br>->Player exposed to socialisation scenarios | NA | [66] |

**Table 2.** *Cont.*

| Game Name | Platform | Game Genre | Player Perspective | Game Dimensionality | Description of Gameplay | Game Mechanics | Control Group's Game | Citation |
|---|---|---|---|---|---|---|---|---|
| Not given | Customised | Social VR | Immersive | 3D (VR) | A blue room was used to project a virtual environment all around the participant and different scenes of different social situations were played with virtual interactring agents. | ->No real reward mechanism ->Player exposed to socialisation scenarios | NA | [68] |

**Table 3.** Participant information of the selected studies.

| Number of Participants | Mean Age | Participant Age Min | Participant Age Max | Type of Affective Disorder | ASD/Neuro | Control Group's Intervention | Citation |
|---|---|---|---|---|---|---|---|
| 37 | 40 | 25 | 55 | Major Depressive Disorder | Neurotypical | Were administered a different game | [98] |
| 56 | 22 | 18 | 26 | Subthreshold or Mild depression | Neurotypical | No intervention | [33] |
| 38 | 9.5 | 7 | 12 | Social Anxiety Disorder | Neurotypical | Were administered SET-C | [34] |
| 117 | 12 | 8 | 16 | Anxiety | Neurotypical | Therapist led Cognitive Behavioural Therapy. | [35] |
| 40 | 13.5 | 10 | 17 | Emotional dysregulation of anger and aggression | Neurotypical | ACT with sham video game | [36] |
| 34 | 60 | 45 | 75 | Major Depressive Disorder | Neurotypical | No control groups. | [37] |
| 77 | 62.5 | 40 | 85 | Depression & Anxiety | Neurotypical | Traditional Cardiac Rehabilitation | [38] |
| 50 | 39 | 27 | 51 | Depressive Disorder, Bipolar Disorder, Anxiety Disorder | Neurotypical | Standard relaxation exercises | [39] |
| 40 | 34 | 27 | 41 | PTSD | Neurotypical | Standard PTSD therapy | [40] |
| 20 | 65 | 53 | 77 | Anxiety | Neurotypical | Control intervention was a tablet-based game with comparable audio, visual, and tactile components | [41] |

Table 3. *Cont.*

| Number of Participants | Mean Age | Participant Age Min | Participant Age Max | Type of Affective Disorder | ASD/Neuro | Control Group's Intervention | Citation |
|---|---|---|---|---|---|---|---|
| 41 | 47.5 | 35 | 60 | Bipolar Disorder | Neurotypical | Treatment as usual | [42] |
| 27 | 14.5 | 13 | 16 | Public Speaking Anxiety (PSA) | Neurotypical | NA | [43] |
| 49 | 18 | 18 | 18 | Recurring depression | Neurotypical | Regular antidepressant drug | [44] |
| 71 | 31 | 19 | 43 | PTSD | Neurotypical | NA | [45] |
| 100 | 34 | 24 | 44 | Spider Phobia | Neurotypical | One session treatment | [46] |
| 30 | 23.5 | 14 | 33 | Dental Phobia | Neurotypical | Informational Pamphlet condition | [47] |
| 1 | 15 | 15 | 15 | Anxiety | Neurotypical | NA | [48] |
| 43 | 10 | 8 | 12 | Anxiety | Neurotypical | Standard therapy | [49] |
| 47 | 71 | 64 | 78 | Depression and Fear of Falling | Neurotypical | NA | [50] |
| 30 | 47.5 | 37 | 58 | Anxiety and depression | Neurotypical | Usual therapy | [51] |
| 24 | 46 | 23 | 69 | Anxiety and depression | Both | Same, just that the 2nd phase trail is considered the control group | [52] |
| 33 | 25.5 | 15 | 36 | Anxiety | Neurotypical | Same, just controlgroup does not have claustrophobia | [53] |
| 180 | 41.5 | 18 | 65 | Acrophobia | Neurotypical | Same, just that the 2nd phase trail with a different is considered the control group | [54] |
| 174 | 9.5 | 7 | 12 | Anxiety | Neurotypical | Coping Cat | [55] |
| 22 | 68 | 62 | 74 | Late life depression | Neurotypical | Problem solving Therapy | [56] |
| 16 | 72 | 60 | 84 | Fear of Falling | Neurotypical | Usual therapy | [57] |
| 63 | 41.5 | 18 | 65 | Anxiety | Neurotypical | Same, just that the2nd phase trail is considered the control group | [58] |
| 15 | 16 | 14 | 18 | Depression | Neurotypical | NA | [59] |
| 72 | 69 | 65 | 73 | Depression | Neurotypical | A different game was given. | [60] |
| 19 | 78.5 | 63 | 94 | Subsyndromal depression | Neurotypical | NA | [61] |
| 8 | 10 | 8 | 12 | Anxiety | ASD | NA | [62] |
| 3 | 8 | 6 | 10 | Anxiety | ASD | NA | [63] |
| 109 | 12 | 8 | 16 | Anxiety | ASD | Another puzzle game | [64] |
| 40 | 11.5 | 8 | 15 | Dental Anxiety | ASD | NA | [65] |
| 9 | 10 | 7 | 13 | Anxiety & Phobia/fear | ASD | NA | [66] |
| 3 | 8 | 6 | 10 | Anxiety | ASD | NA | [67] |
| 8 | 37.5 | 18 | 57 | Phobia | ASD | NA | [68] |

**Table 4.** Serious game intervention methodologies and outcomes.

| Intervention | Evaluation Technique | Outcome | Experimental Methodology | Data Collection Technique | Citation |
|---|---|---|---|---|---|
| Administered AKL-T03 through a videogame. | Analysis of Covariance (ANCOVA) | Significant improvement in sustained attention and in cognitive functioning. | Double-blind randomized controlled trial | Realtime with remote servers and some post-game questionnaires | [98] |
| Music-based casual video game. | Depression anxiety and stress scale (DASS-21), Positive and Negative Affect Scale (PNAS), General Self-efficacy Scale (GSES), Emotion Regulation Questionnaire(ERQ) | The depression, anxiety, and stress symptoms were significantly reduced in the experimental group participants after 4 weeks of music-based video game training compared with the control group. | Randomized controlled trial | Realtime through the game, pre- and some post-game questionnaires | [33] |
| Social VR game | Clinical Global Impressions Scale (CGI), Children's Global Assessment Scale (C-GAS), Child Behavior Checklist (CBCL), Social Phobia and Anxiety Inventory for Children-Parent Version (SPAIC-PV) | Decreased anxiety and improved social skill in social encounters. | Randomized controlled trial | Observation, pre- and some post-game questionnaires | [34] |
| An adventure, third-person neurofeedback game. | Spence Children's Anxiety Scale (SCAS), State Anxiety Measure, Heart Rate | All measures of anxiety significantly decreased. | Randomized controlled trial | Observation, during a game session by pausing gameplay and some post-game questionnaires | [35] |
| A 'Space Invaders' themed game was administered. | State-Trait Anger Expression Inventory-Child Adolescent (STAXi-CA), Modified Overt Aggression Scale (MOAS), Disruptive Behavior Rating Disorder Scale (DBDRS), Clinical Global Impressions Severity/Improvement (CGI-S and CGI-I) | Reduced behavioral expression of anger, but not the experience of angry feelings. | Double-blind randomized controlled trial | Pre and post-game questionnaires | [36] |
| Racing Game (AKL-T01) was administered. | fMRI during a Stroop/Flanker task, resting-state functional connectivity (rsFC), attending or ignore distractor (AID), Conners' continuous performance Test (CPT), self-reported subscale of Frontal systems behavior scale (FrSBe), Patient Health Questionnaire (PHQ-9) | A significant reduction in depressive mood symptoms | Two-phase clinical trial | Pre and post-game questionnaires and brains scans during gameplay | [37] |

**Table 4.** *Cont.*

| Intervention | Evaluation Technique | Outcome | Experimental Methodology | Data Collection Technique | Citation |
|---|---|---|---|---|---|
| A Therapeutic VR game was administered. | Hospital Anxiety and depression Scale (HADS), Perception of Stress Questionnaire (PSQ) | Reduced the level of anxiety and depression symptoms. | Randomized controlled trial with a blinded outcome assessor | Post-game questionnaires | [38] |
| A Therapeutic VR game was administered. | Visual Analog Scales (VAS), Perceived Stress Scale, Inventory of Depressive Symptomatology-Self-Report, Simulator Sickness Questionnaire | Intervention had a stronger beneficial effect on momentary anxiety, sadness, and cheerfulness. | Randomized crossover trial | Post-game questionnaires | [39] |
| EMDR Tetris video game. | Posttraumatic Diagnostic Scale (PDS), Beck depression Inventory II (BDI-II), State-Trait Anxiety Inventory (STAI), MRI scanning | Hippocampal volume increased in the Tetris group, indicating reductions in symptoms of PTSD, depression and anxiety. | Controlled trial | Post-game questionnaires and MRI scans | [40] |
| An action VR game was administered. | State-Trait Anxiety Inventory | Significant reductions in feeling tense, strained, and upset. | Single blinded randomized controlled trial | Post-game questionnaires | [41] |
| A social simulation game where the player avatar has Bipolar Disorder was administered. | Medication Adherence Rating Scale (MARS), Drug Attitude Inventory (DAI-10) | Intervention may help patients with Bipolar Disorder to increase their confidence in medications, if used regularly. | Two-arm open randomized controlled trial | Pre and post-game and follow-up questionnaires | [42] |
| A social VR game was administered that encourage adolescents to communicate with classmates. | Heart rate, Public Speaking Anxiety Scale (PSAS), Social Interaction Anxiety Scale (SIAS) | Significant decrease in PSA symptoms. | Non-randomized feasibility and pilot trial | Pre and post game and follow-up questionnaires | [43] |
| Administered an off-the-shelf videogame called Plants vs. Zombies. | Patient Health Questionnaire (PHQ-9) | Treatment-resistant depression symptoms (TRDS) and improving heart rate variability (HRV). | Non-randomized controlled trial | Pre and post-game and follow-up questionnaires | [44] |
| Tetris video game and physical exercise was administered. | Depression Anxiety Stress Scales short form (DASS-21) | No significantly different in PTSD symptoms were found for the intervention. | Unblinded two-phased trial | Pre and post game and follow-up questionnaires | [45] |

**Table 4.** *Cont.*

| Intervention | Evaluation Technique | Outcome | Experimental Methodology | Data Collection Technique | Citation |
|---|---|---|---|---|---|
| A Therapeutic VR game was administered. | Behavioral Approach Test (BAT), Structured Clinical Interview for DSM-IV Axis-I Disorders (SCID-I/P), Spider Phobia Questionnaire (SPQ), Generalized Anxiety Disorder Assessment (GAD-7), Negative Effects Questionnaire (NEQ-32), Igroup Presence Questionnaire (IPQ) | Stronger reductions behavioral avoidance and self-reported fear in the intervention group than the control. | Parallel group randomized non-inferiority trial | Pre and post game and follow-up questionnaires | [46] |
| A Therapeutic VR game was administered. | Visual Analogue Scale-Assessment (VAS-A), Modified Dental Anxiety Scale (MDAS), Dental Fear Survey(DFS), Behavioural Avoidance Test (BAT) | Showed a significant reduction in anxiety scores. | Single-blind randomized controlled trial | Pre and post game and follow-up questionnaires | [47] |
| A Therapeutic VR game was administered. | State-Trait Anxiety Inventory for Children (STAIC), Positive and Negative Affective Scale (PANAS), Resilience Scale, Avoidance and Fusion Questionnaire for Youth (AFQ-Y), The Willingness and Action Measure for Children and Adolescents (WAM-C/A) | Results show good acceptability and feasibility, improved state and trait anxiety, resilience, and emotional competence in controlling behavior. | Case study | Pre and post game and follow-up questionnaires | [48] |
| An adventure, third person neurofeedback game. | Spence Children's Anxiety Scale (SCAS-C), Diagnostic and Statistical Manual of Mental Disorders(DSM-IV) | Results showed that changes in in-game play behaviours representing therapeutic exposure techniques predicted improvements in anxiety symptoms 3 months later. | Randomized controlled trial | Video capture of player reactions as well as pre and post game questionnaires | [49] |
| An exergame was administered. | Geriatric Depression Scale (GDS), Falls Efficacy Scale-International Brazil (FES-I Brazil), Biodex System 4 Dynamometer | Indicated to decrease depressive symptoms in fallers and increase the peak torque in non-fallers among community-dwelling older women. | Non-randomized pilot trial | Post-game questionnaire and during game observation | [50] |

**Table 4.** *Cont.*

| Intervention | Evaluation Technique | Outcome | Experimental Methodology | Data Collection Technique | Citation |
|---|---|---|---|---|---|
| A mobile shooting game was administered. | Beck depression Inventory (BDI), Beck Anxiety Inventory (BAI), Stress Response Inventory (SRI) | BDI and SRI scores in the Game group greatly decreased compared with those in the Nongame group. | Randomized controlled trial | Pre and post game and follow-up questionnaires | [51] |
| An adventure, third person game was administered. | Novel interview schedule | Improved symptoms but not significantly. | Pilot randomized controlled trial | Post-game questionnaires | [52] |
| An exploration oriented third person game was administered. | Novel interview schedule | Anxiety measure showed a decrease. | Controlled trial | Pre and post-game questionnaires | [53] |
| A customised Exploration game was administrated. | Acrophobia Questionnaire, Attitudes Towards Heights Questionnaire (ATHQ), Beck Anxiety Inventory (BAI), System Usability Scale (SUS), Igroup Presence Questionnaire (IPQ), Patient Health Questionnaire (PHQ-9) | Anxiety measure showed a decrease. | Two phase randomized controlled trial | Pre and post game and follow-up questionnaires | [54] |
| An adventure, third person neurofeedback game. | Spence Children's Anxiety Scale (SCAS) | Decreased anxiety more than control intervention. | Randomized controlled non-inferiority trial | Pre and post game and follow-up questionnaires | [55] |
| Racing Game (AKL-T01) was administered. | Hamilton Depression Rating Scale (HAM-D), Patient Health Questionnaire (PHQ-9) | Mood is improved compared to control. | Proof-of-concept trial | Pre and post game and follow-up questionnaires | [56] |
| A Therapeutic VR game was administered. | Spielberger State–Trait Anxiety Inventory (state anxiety: STAI form Y-A; trait anxiety: STAI form Y-B), Sheehan Disability Scale (SDS), Beck depression Inventory (BDI) | No significantly different effect on depressive symptoms were found for the intervention. | Randomized controlled trial | Pre and post-game questionnaires | [57] |
| An aggregation of mini games was administered. | Anova | No significantly different effect on anxiety and panic symptoms were found for the intervention. | Unblinded parallal group randomized trial | Pre and post game and follow-up questionnaires | [58] |

**Table 4.** *Cont.*

| Intervention | Evaluation Technique | Outcome | Experimental Methodology | Data Collection Technique | Citation |
|---|---|---|---|---|---|
| A customsed social interaction game was administered. | Acceptability score, Custom developed questionnaire for psychometrics | Acceptability score and other measures suggest such a game could be useful in decreasing depressive mood. | Pilot trial | Pre and post-game questionnaires | [59] |
| A custom developed collection of games were administered. | Profile of Mood State second edition (POMS2), World Health Organization Subjective Well-being Inventory (WHO-SUBI) | Reduced depressive mood more than control. | Single-blinded randomized control trial | Pre and post-game questionnaires | [60] |
| A collection of exergames were administered | Depressive Symptoms (QIDS), Beck Anxiety Inventory (BAI) | Significant reduction in depressive symptoms. | Pilot trial | Pre and post-game questionnaires | [61] |
| An adventure, third person neurofeedback game. | Spence Children's Anxiety Scale (SCAS), Spence Child Anxiety Scale for Parents (SCAS-P), Anxiety Disorders Interview Schedule for DSM-IV | The effect MindLight has on anxiety reduction is not enhanced by concurrent CBT. | Pilot trial | Pre and post-game questionnaires | [62] |
| A custom appplication of a game and a control panel for parents were administered. | Spence's Children Anxiety Scale (SCAS), Spence Children Anxiety Scale-Parents (SCAS-P) | Not significant results as the participants did not all follow the protocols. | Pilot trial | Pre and post-game questionnaires accompanied with interviews. | [63] |
| An adventure, third person neurofeedback game. | Spence Children's Anxiety Scale (SCAS), Spence Child Anxiety Scale for Parents (SCAS-P) | Not significant results. | Randomized controlled trial | Pre and post-game questionnaires | [64] |
| A customised VR environment was administered. | Venham's Picture Test, Frankel's Behavior Rating scales | Significant reduction in anxiety symptoms. | Pilot trial | Pre and post game and follow-up questionnaires | [65] |
| A Therapeutic VR game was administered | Spence Children's Anxiety Scale-parent version (SCAS-P) and child version (SCAS-C), Target Behaviours scale | Merging CBT and VR proved to be highly effective. | Pilot trial | Pre and post-game questionnaires | [66] |

**Table 4.** *Cont.*

| Intervention | Evaluation Technique | Outcome | Experimental Methodology | Data Collection Technique | Citation |
|---|---|---|---|---|---|
| A custom appplication of a game and a control panel for parents was administered. | Spence's Children Anxiety Scale (SCAS), Spence Children Anxiety Scale-Parents (SCAS-P) | Not significant results as the participants did not all follow the protocols. | Pilot trial | Pre and post-game questionnaires accompanied with interviews. | [67] |
| A Therapeutic VR game was administered | Spence Children's Anxiety Scale-parent version (SCAS-P) and child version (SCAS-C), Target Behaviours scale, Beck Anxiety Inventory (BAI), Generalized Anxiety Disorder 7 (GAD-7), Patient Health Questionnaire-9 (PHQ-9) | Merging CBT and VR proved to be highly effective. | Pilot trial | Pre and post-game questionnaires | [68] |

### 3. Observations and Analysis

Out of the four scientific databases used, Embase provided the most relevant sets of articles and so approximately 11.8% of its search results were present in the final selection of articles, followed by 7% for Scopus, 4.2% for Web Of Science and 0% for IEEE Xplore. Most of the search results from IEEE and some from Web Of Science were about digital and non-digital interventions for autistic individuals who either did not have any affective disorders or only had comorbidities for those. However, during the abstract screening, it was found that all of the articles from IEEE were either study protocols or revisions of some studies that had been published before. Hence, no articles extracted from IEEE were included in the final list of articles to be reviewed.

#### 3.1. Characteristics of Selected Studies

The selected articles were written in English and were published between the years 2008 and 2022 (as no related publication was found in 2023 in the search results). The publications with the highest number of articles were 'Games for Health Journal' (4) followed by the 'Journal of Behavior Therapy and Experimental Psychiatry' (2). In terms of publication years, 2019 and 2022 saw the highest number of articles published at 9 and the second highest number of articles published was in 2018 at 7. In total, 34 out of 37 articles were journal papers and only 3 were conference papers. Country-wise, the USA produced the highest number of articles (9 out of 37) closely followed by the Netherlands (7 out of 37). In a continental comparison, Europe led the publications list with 16 articles that were published between 2016 and 2022, followed by North America with 9 articles that were published between 2010 and 2022. Tables 2–4 show a more complete picture of the serious games utilized, the population details of the participants and the intervention details of the selected studies.

The population in the selected articles mostly consisted of neurotypical individuals, being present in nearly 78% of the articles, while autistic individuals were present in nearly 19% of the articles and 3% of the articles were about both populations. The mean age of the participants was between 8 and 78.5. The mean age of the neurotypical participants was 35.5 and for the ASD participants it was 13.9. For 7 out of 37 studies, the mean age of the participants was above 45, meaning most of the studies were focused on the younger population of ASD and neurotypical individuals.

The most commonly utilized experimental methodology was 'pilot study' at eight articles, closely followed by 'randomized controlled study' at seven articles. Out of the 37 selected articles, 8 used games on portable platforms (iPad OS, iOS, Android, etc.) [37,39,41,43,46,51,52,54,56,58,60,63,67,98], 6 PC games (Windows) [47–49,52,53,57], 2 used console games (Nintendo DS XL and Nintendo's Wii Sports) [40,61], 2 were web-based [42,59] and 5 utilized custom platforms [33,38,65,66,68]. In most cases, the control group was given standard therapy (7 out of 37), closely followed by being given a different game than the intervention (5 out of 37). Several studies provided different alternative support or therapies to the control groups; for instance, 9 studies provided standard therapy for the respective affective disorder to the control groups [35,38–40,42,44,47,51,55] and 10 were given either a different, the same game with some changes or the same game at phase 2 trials [34,36,41,52–54,58,60,62,64,98]. This, along with the heterogeneity of the population in the studies, meant that it was not possible to carry out a further meta-analysis of the studies' findings.

The most prevalent affective disorder in the selected literature was anxiety. Anxiety was the sole focus of 11 articles, was accompanied by other disorders in 4 other articles and in 2 other articles was included as social or dental anxiety. Major depressive disorder and post-traumatic stress disorder had the second highest prevalence at 2 articles each. However, depression was included in 11 articles combined with other disorders, indicating a higher comorbidity of depression with other affective disorders.

When it comes to game genres, there are several schools of thought on how games should be classified based on characteristics including storyline and dynamics [94,99]. Some of the games encountered in this review did not fit the traditional genres; for instance, the most frequently occurring game genre in the articles were VR games that were purpose-built for relaxation and distraction by just moving around virtual worlds or just conversing with VR agents. Hence, we coined the genre terms "therapeutic VR" and "social VR" respectively for these games. Therapeutic VR was found in the highest number of articles at 6 out of 37, followed by social VR at 4 out of 37, which was closely followed by racing games and puzzle games at 3 articles each. This indicated that for certain affective disorders, simple VR implementations could bring about largely beneficial outcomes for the patients. Furthermore, as VR games have a higher level of immersion than traditional 3D games played on a flat screen [100], we opted to use 'immersive' as the player perspective for VR games and used 3D (VR) as their dimensionality. A total of 8 out of the 37 studies did not disclose the names of the games they employed while the rest mentioned one game, an application package with a game and control center, or a suite of several mini games. A probable reason for the former could be that the unmentioned games were bespoke for the study and were too simplistic, for instance, a therapeutic VR where the dynamics were very limited and the player could only walk through several virtual worlds.

### 3.2. Methodological Characteristics of Selected Studies

Table 4 shows the experimental methodologies that were employed in the studies. In total, 22 different experimental methodologies were used, which included the following: double-blind randomized controlled trial, randomized controlled trial, two-phase clinical trial, randomized controlled trial with a blinded outcome assessor, randomized crossover trial, controlled trial, single-blinded randomized controlled trial, two-arm open randomized controlled trial, non-randomized feasibility and pilot trial, non-randomized controlled trial, unblinded two-phased trial, parallel-group randomized non-inferiority trial, single-blind randomized controlled trial, case study, non-randomized pilot trial, pilot randomized controlled trial, two-phase randomized controlled trial, randomized controlled non-inferiority trial, proof-of-concept trial, unblinded parallel-group randomized trial, pilot trial, and single-blinded randomized control trial. There were 17 studies (45.9%) that had some form of control groups to compare their findings with, while 20 (54.1%) involved some form of randomization in the subject allocation to the experiment group, indicating that both techniques were favored by the authors of the reviewed studies more than others, such as pilot studies or case studies.

Table 4 also shows the evaluation technique or measures that were used in the studies in different combinations to assess the impact of the CSG interventions. There were 30 different evaluation technique/measures among the studies, which included analysis of covariance, depression anxiety and stress scale (DASS-21), clinical global impressions scale (CGI), Spence children's anxiety scale (SCAS), state-trait anger expression inventory—child and adolescent (STAXi-CA), fMRI during a stroop/flanker task, hospital anxiety and depression scale (HADS), visual analog scales (VAS), post-traumatic diagnostic scale (PDS), state-trait anxiety inventory, medication adherence rating scale (MARS), heart rate, patient health questionnaire (PHQ-9), behavioral approach test (BAT), visual analogue scale assessment (VAS-A), state-trait anxiety inventory for children (STAIC), Spence children's anxiety scale (SCAS-C), geriatric depression scale (GDS), beck depression inventory (BDI), novel interview schedule, novel interview schedule, acrophobia questionnaire, Spence children's anxiety scale (SCAS), Hamilton depression rating scale (HAM-D), acceptability score, profile of mood state second edition (POMS2), depressive symptoms (QIDS), Venham's picture test', and Spence children's anxiety scale parent version (SCAS-P) and child version (SCAS-C). The most frequently used technique was Spence children's anxiety scale (SCAS) in nine studies for anxiety and its comorbidities [35,49,55,62–64,66–68] followed by patient health questionnaire (PHQ-9) in six studies for depression and its comorbiditie [37,44,54,56,66,68].

There were 12 data collection techniques employed which were mostly similar to each other. These included real-time with remote servers and some post-game questionnaires, real-time through the game, pre- and some post-game questionnaires, observation, pre- and some post-game questionnaires, observation, during a game session by pausing gameplay and some post-game questionnaires, pre- and post-game questionnaires, pre- and post-game questionnaires and brains scans during gameplay, post-game questionnaires post-game questionnaires and MRI scans, pre- and post-game and follow-up questionnaires, video capture of player reactions as well as pre- and post-game questionnaires, post-game questionnaire and during game observation, pre- and post-game questionnaires accompanied with interviews. Interestingly, the most frequently used data collection technique involved follow-up questionnaires (35%) indicating that a fair number of studies reviewed rigorous investigation of the effectiveness of the application of serious games for affective disorders. The most frequently used technique was pre- and post-game and follow-up questionnaires at 13 studies (35%) followed by pre- and post-game (27%) which makes sense as this indicates that authors of most of the studies opted to take pre-game measurements to compare them to post-game ones. However, only 35% of the studies carried out a follow-up, which is much higher than reported in other reviews [69,73,74], but still indicates that in most cases, follow-up studies were not conducted.

Table 4 shows the outcomes of the studies reviewed in this research. In total, 81.1% of the studies reviewed showed a positive impact of serious game interventions for autistic or neurotypical individuals with affective disorders, indicating that such games could be a viable tool for therapeutic purposes. Some possible explanations for the studies that did not find a positive impact of such applications have been provided in Section 3.4.

### 3.3. Serious Games for Affective Disorders

There were five different types of affective disorders that were investigated in the selected studies. Almost all of them were investigated solely as well as being comorbid with other affective disorders and their distributions among the selected studies are shown in Figure 2. Table 2 shows the names and characteristic details of the games that had been applied as a therapeutic intervention or to mediate therapeutic interventions for autistic or neurotypical patients with affective disorders in the reviewed studies. There were 21 different games utilized in the reviewed studies, which were 'MindLight', 'Neuro Racer', 'Pegasas VR', 'ACT RAGE-Control', 'Vrelax', 'Tetris', 'Bear Blast', 'BIPOLIFE', 'Plants vs. Zombie', 'VIMSE', 'Space Academy', 'Dance Central', 'Hit the Cancer', 'Pesky Gnats: The Feel Good Island', 'Claustrophobia Game', '0Phobia', 'Flowy', 'Maya', 'Processing Speed Training Game', 'Wii Sports Gaming Package', 'New Horizon and SpaceControl'. The most frequently used game was a neurofeedback third-person adventure game called 'MindLight' (5 out of 37) [35,49,55,62,64] and this was also the most often used game for anxiety therapy, among others such as 'New Horizon and Space Control' (2 out 37) [63,67]. There were some other games as well for other forms of anxiety, for instance, the puzzle game 'Tetris' was used for PTSD symptom mitigation [40,45] and the therapeutic VR game 'Pegasas VR' was used for therapy of social anxiety disorder [34]. Moreover, there were other games that were used for therapy when anxiety was comorbid with other affective disorders, for instance, the therapeutic VR game 'Vrelax' for therapy for depressive, bipolar and anxiety disorders [39], the third person adventure game 'Pesky Gnats: The Feel Good Island' [52] and the first person shooter game 'Hit the Cancer' [51] for anxiety and depression. Among these games, 'MindLight' [62,64], 'New Horizon and Space Control' [63,67] and three other unnamed games [65,66,68] were used for interventions for participants with ASD while the rest were utilized as interventions for neurotypical participants [33–49,51,53–61,98]. Among these games, 'Hit The Cancer' [51] and 'Dance Central' [50] were utilized in two serious game interventions where the participants had comorbidity of breast cancer (for the former) and musculoskeletal function dysfunction (for the latter); the interventions of the games were solely aimed at the respective affective disorders of those participants.

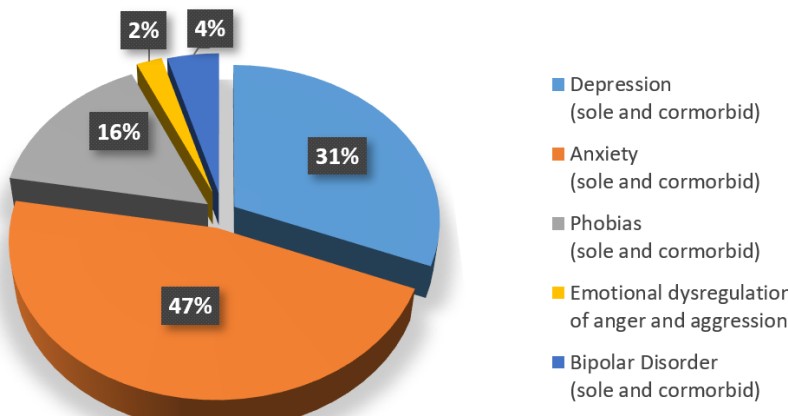

**Figure 2.** Distribution of affective disorders that were investigated by the selected studies.

Four different games were utilized for therapy against different types of phobias, two of them were named in the studies, namely the therapeutic VR game 'VIMSE' for spider phobia [46] and the action-exploration VR game '0Phobia' for acrophobia [54]. Three games were utilized for phobias/fear disorders being comorbid with other affective disorders and the only game that was comorbid with an ailment other than an affective disorder among these was the exergame 'Dance Central' (for depression and fear of falling) [50].

The most frequently used game for depression therapy was the racing game 'Neuro Racer' (3 out of 37) [37,56,98] followed by a first-person role-playing adventure game 'Maya' [59] and a set of puzzle games 'Processing Speed Training Games' [60]. Some other games were utilized for some forms of depression, for instance, the exergame 'Wii Sports Gaming Package' for subsyndromal depression [61] and the action-shooter game 'Plants vs. Zombies' for recurring depression [44]. In the studies reviewed for this paper, depression was found to be comorbid with anxiety, phobias, and one instance of bipolar disorder. The games utilized for those studies have already been discussed above.

There was only one study that investigated therapy with a serious game solely for bipolar disorder, which utilized the web-based social simulation 'BIPOLIFE' [42]. The only other disorder for which a serious game intervention was investigated among the selected studies was emotion dysregulation of anger and aggression; the game utilized for that study was the isometric action game 'ACT RAGE-Control' [36]. Table 2 shows the details of the data collected from the selected articles on the games utilized for affective disorder therapies.

From Table 2, there were 11 game genres, which included 'racing', 'music', 'social VR', 'adventure', 'action', 'therapeutic VR', 'puzzle', 'social simulation','shooter', 'exergame' and 'exploration'; there were 4 player perspectives which included 'first person', 'isometric', 'immersive' and 'third person' and there were 3 game dimensionalities, which were 2D, 3D and 3D (VR). The most frequently used game genre was adventure (21%), followed by therapeutic VR (16.2%). Immersive was the most frequently used player perspective (35%) followed by first-person and isometric (21.6%). Finally, the most frequently used game dimensionality was 3D (VR) (43.2%), followed by 3D (35%) indicating games with immersive environments were favored for the studies by the respective authors.

In terms of mechanics, five games did not involve any reward mechanisms [34,43,47,57,66,68], half of which were for anxiety and the other half for phobias. However, the rest of the 16 games used some form of reward scoring, which indicates that having reward mechanisms in serious games for therapeutics for affective disorders can be beneficial.

### 3.4. Impact of Interventions on Affective Disorders

Out of the 37 studies reviewed, 7 showed no significant impact of the serious game intervention as compared to the control group. The affective disorders for these were

either anxiety (5 out of 37; 4 ASD and 1 neurotypical) [58,62–64,67], PTSD (1 out of 37; 1 neurotypical) [45] and fear of falling (1 out of 37; 1 neurotypical) [57]. There were no obvious similarities between these studies and hence no conclusive reasoning could be synthesized for the lack of positive impact from serious game interventions for these studies, apart from the following:

- Both [62,64] mention that MindLight and CBT interventions were effective on their own but together they do not have any additional positive impact on the outcome.
- Both [63,67] mention that their study protocols were not entirely observed by the participants and hence the findings could not be conclusive.

However, there was ample evidence that the serious game interventions in the rest of the 30 studies did produce a net positive impact as a therapy tool or as a mediator to associated therapies for affective disorders. In total, 3 of these studies were for the autistic population [65,66,68] while the other 27 were for the neurotypical population. In fact, the authors of [66,68] argued that these two together could be highly effective in reducing patients' phobias, which is in contrast to what the authors of [62,64] found about applying serious games with CBT for anxiety therapy, as often phobias are classified as anxiety disorders [101].

There were 4 studies that had more than or equal to 100 participants among the 37 selected studies, which were 180 for the study that employed phobia as a therapy for acrophobia [54], 174 for the study that employed MindLight as a therapy for anxiety [55], 117 for the study that also employed MindLight as a therapy for anxiety [35] and 100 for the study that employed VIMSE as a therapy for spider phobia; all of them showed a positive impact of the serious game intervention on their participants.

Games such as MindLight seem to be most effective for the younger population as a sole anxiety therapy, as was evident from the relatively large populations for the studies [35,49,55] where the mean ages of the participants were 12 and 9.5, respectively. On the other hand, several VR games showed a significant positive impact as therapies for different types of anxieties, phobias and depression [34,38,39,41,43,46–48,54,65,66,68] and a majority of these cases employed a therapeutic VR with considerably good outcomes. Table 4 provides greater details on the methodology, data collection techniques and interventions that were employed in the studies along with their outcomes.

## 4. Results and Discussion

In order to present the findings of our analysis of the selected articles, we present what we learned from them with respect to our research questions for this study as follows:

1. Can serious games be effective as therapies for affective disorders in autistic and neurotypical individuals?
   There were 30 different outcome measures utilized in different combinations in the studies and eight of them showed no significant impact of serious game interventions in seven studies [45,57,58,62–64,67]; however, apart from 1, 7 of these outcome measures and 22 of the other outcome measures in the rest of 30 studies showed a positive impact of the serious games intervention as a therapy or mediator of other therapies for different affective disorders for both ASD and neurotypical populations. Therefore, the studies support the application of serious games for affective disorders in autistic and neurotypical individuals. Even though several of the studies did not have the appropriate sample sizes for their results to be statistically significant, for instance, 9 out of 37 studies had sample sizes less than 20, still more than 75% of the studies we reviewed had an appropriate sample size and hence this further solidifies the applicability of serious games for affective disorders in autistic and neurotypical individuals.
2. For which affective disorders among autistic or neurotypical individuals have serious games been effectively applied?
   Table 3 shows the details of the participants in the studies reviewed. In general, there were five different affective disorders found in the populations of the studies, which

were depression, anxiety, phobias, bipolar disorder and their comorbidities, along with emotional dysregulation of anger and aggression. Together with Tables 3 and 4, it can be seen that there were successful applications of serious games for all of these affective disorders in at least one of the studies reviewed. Depression was shown to have the highest comorbidity, appearing with other affective disorders in five studies [38,39,50–52], followed by anxiety appearing in four studies [39,51,52,66].

3.  What are the design characteristics of the games that are effective for different affective disorders?

The charts in Figures 3–5 present the characteristics of the games that were used effectively for depression (and its comorbidities), anxiety (and its comorbidities), phobias (and its comorbidities), bipolar disorder (and its comorbidities) and emotional dysregulation of anger and aggression in the selected studies. Figure 3 shows the distribution of games in terms of the different game dimensionalities observed (2D, 3D or 3D (VR)), Figure 4 shows the distribution of games in terms of the different player perspectives in the game that were observed (first person, third person, immersive or isometric) and finally Figure 5 shows the distribution of games in terms of the different game genres observed (adventure, driving, puzzle, shooter, etc.)

From Figure 3, it is apparent that 3D games were more frequently utilized for intervention for depression patients, which was 3D (VR) and 3D for anxiety, predominantly 3D (VR) for phobias, 3D (VR) and 3D for bipolar disorder and 2D for emotional dysregulation of anger and aggression in the selected studies. Figure 4 clearly shows that first-person and immersive games were more frequently utilized for intervention for depression patients, which was immersive and third-person for anxiety, again predominantly immersive for phobias, isometric for bipolar disorder as well as for emotional dysregulation of anger and aggression in the selected studies. As isometric games can have 2D, 3D, or even a hybrid of these two dimensionalities, and hence are very versatile in terms of implementation [102], it is understandable that such games were chosen for interventions of two different affective disorders. It can be seen in Figure 5 that racing games have been utilized the most, followed by therapeutic VR, exergames and shooter games for intervention for individuals with depression. For interventions for individuals with anxiety, adventure games and social VR were utilized the most, followed by therapeutic VR. Therapeutic VR and social VR were the most utilized for intervention for individuals with phobias, followed by racing and exploration games. For individuals with bipolar disorder, therapeutic VR and social simulation games were mostly utilized as interventions, and for emotional dysregulation of anger and aggression an action game was utilized.

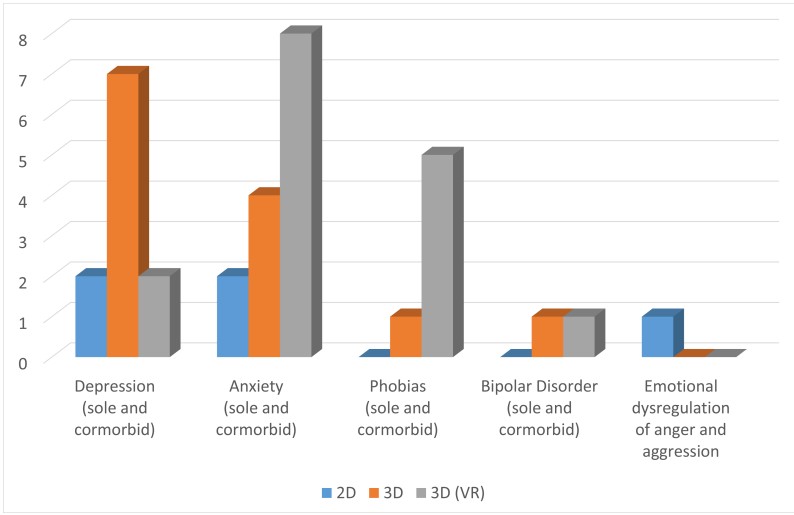

**Figure 3.** Dimensionality of the games used for each affective disorder.

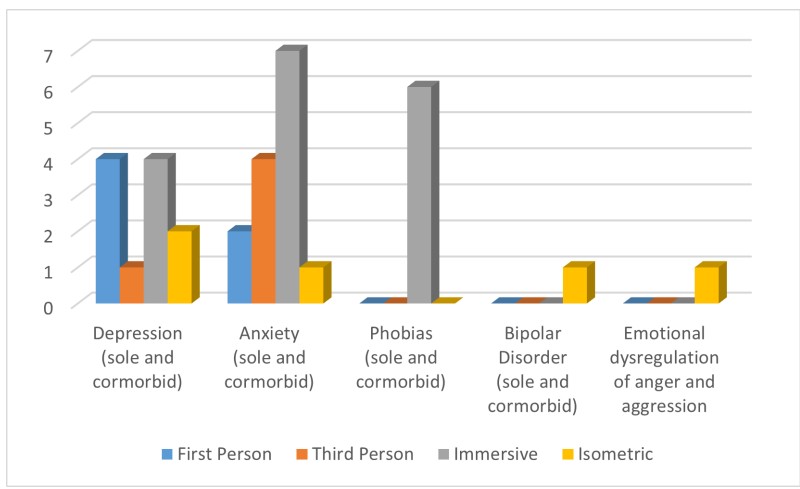

**Figure 4.** Player perspectives of the games used for each affective disorder.

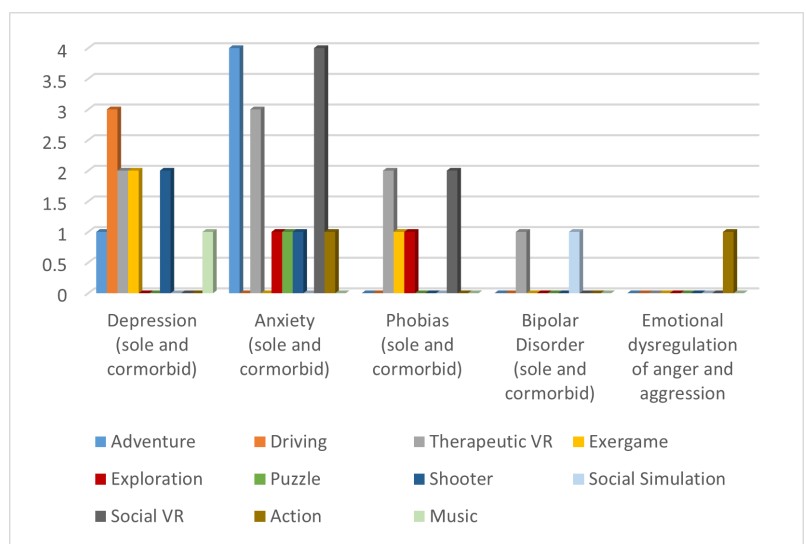

**Figure 5.** Genre of the games used for each affective disorder.

Figures 3–5 clearly show some trends for the game design characteristics that had been applied in games which were effectively utilized as therapies for the different affective disorders, which were:

- For depression and its comorbidities: 11 games were applied and the most frequently occurring player perspective, game dimensionality and genre were 3D at 63.64% (7/11), first person/immersive at 36.36% (4/11) each and racer at 27.27% (3/11), respectively.
- For anxiety and its comorbidities: 14 games were applied and the most frequently occurring player perspective, game dimensionality and genre were 3D (VR) at 57.14% (8/14), immersive at 50% (7/14) and adventure at 28.57% (4/14), respectively.
- For phobias and its comorbidities: six games were applied and the most frequently occurring player perspective, game dimensionality and genre were 3D (VR) at 83.33% (5/6), immersive at 100% (6/6) and therapeutic VR/social VR at 33.33% (2/6) each, respectively.
- For bipolar disorders and its comorbidities: two games were applied and the most frequently occurring player perspective, game dimensionality and genre were 3D/3D (VR) at 50% (1/2) each, isometric 100% (2/2) and therapeutic VR/social VR at 50% (1/2) each, respectively.

- For emotional dysregulation of anger and aggression: one game was applied and its perspective, game dimensionality and genre were 2D, isometric and action, respectively.

Hence, it could be stated that for depression and its comorbidities, serious games that were 3D, first-person/immersive, and racing games were more effective as therapies in the reviewed studies. Similarly, 3D (VR), immersive, adventure or social VR games have been more effective interventions for anxiety and its comorbidities while 3D (VR), immersive, and therapeutic VR/social VR games have been more effective as interventions for phobias. In the case of serious game applications for bipolar disorder as well as emotional dysregulation of anger and aggression, there were far too few studies that fit the eligibility criteria for this review and so any kind of trends observed for these two affective disorders cannot be free from bias. This is understandable as 3D and 3D (VR) games promote greater levels of better engagement through an increased presence in the virtual world than 2D games [44,103]. Likewise, between first-person and third-person player perspectives, players tend to show greater engagement when they see the game world through the eyes of the playing character [104]; hence, for a 3D game, it makes sense for successful interventions to utilize this fact by utilizing the first person player perspective. As for the genres of games seen in the reviewed studies, racing, adventure and therapeutic/social VR games all provide a distraction from negative thoughts and emotions, relaxation and a sense of accomplishment [44,46,49,52,68,98,105]. Additionally, adventure games have an added benefit of the element of exploration [59], which makes the experience rewarding and hence engaging. An important point to note is that for depression, whether for ASD or the neurotypical population, all instances of serious game interventions showed a positive impact on the participants. These points could lead to a hypothesis that serious games could be applied effectively for depression for both ASD or the neurotypical population, but it will need further research to substantially establish this as a fact.

As for the limitations of our study, our search strategies were only able to provide us with studies involving depression, anxiety, phobias, bipolar disorder and emotion dysregulation of anger and aggression, which are just a handful among a vast variety of different affective disorders [106,107]. This could be due to our search strategies being too focused on the Emtree synonyms, which resulted in only including the most frequently studied affective disorders and did not cover the entire spectrum of other affective disorders. Furthermore, only a few of the studies reviewed attempted to explain why the serious game interventions were effective [33,36–39,47]. A study indicated that being in the 'flow' state induced a sense of control and induced positive emotions in the participants [33]. Others presented the concept of reinforcing positive behaviors through serious game interventions resulting in better anger management [36], and exposure to pleasant, natural scenery in the game environment promoted a positive affective state and reduction in stress levels [39,47]. Furthermore, repeated cognitive stimulation through serious game exposure was indicated to engage the cognitive control network, which could disrupt dysfunctional brain networks, reducing the occurrence of MDD [37]. Since the majority of these studies did not investigate or explain why their serious game interventions were effective and it was not possible to identify common patterns from the ones that did, we could not derive conclusions on the underlying reasons for the serious game interventions effectiveness for the affective disorders investigated in the reviewed studies.

## 5. Conclusions and Future Directions

In this qualitative literature review, we reported several serious game applications as therapy interventions for different affective disorders in both autistic and neurotypical individuals. In 30 out of 37 studies, at least one of the outcome measures showed the positive impact of the serious intervention for the specific affective disorder they investigated. Therefore, we argue that games created for affective disorder therapies are effective and patients and therapists can use the significant benefits of these games to recover from or mediate other therapies for certain affective disorders.

Our analysis, which was reported in Sections 3 and 4, showed that for interventions for depression and its comorbidities, the most frequently utilized game design characteristics (player perspective, game dimensionality and genre, respectively) were 3D at 63.64%, first person/immersive at 36.36%, and racer at 27.27% respectively. For anxiety and its comorbidities, these were 3D (VR) at 57.14%, immersive at 50% and adventure at 28.57% while for phobias and their comorbidities, these were 3D (VR) at 83.33%, immersive at 100% and therapeutic VR/social VR at 33.33% respectively. Therefore, the literature suggests that the implementation of 3D, first person/immersive and racing games could be suitable as interventions for depression and its comorbidities, as well as the implementation of 3D (VR), immersive and adventure or social VR games, could be suitable as interventions for anxiety and its comorbidities and 3D (VR), immersive and therapeutic VR/social VR games could be suitable as interventions for phobias and their comorbidities. Due to a very small number of studies on applications of serious games on bipolar disorder and its comorbidities as well as emotional dysregulation of anger and aggression fitting our eligibility criteria for this study, we could not conclude on the appropriate game design characteristics for applications of serious games as therapies for these affective disorders. To prevent bias, factors such as the variety of criteria and differences in their measurement, small sample size, lack of accurate evaluations of clinical trials on games, etc., needed to be avoided.

Another finding worth highlighting is the fact that, amongst the different affective disorders that had been studied in the reviewed studies, all instances of serious game applications as therapies for depression showed a positive impact on the participants of the related studies in both the ASD and neurotypical populations. Hence, we encourage further investigation of serious game applications as therapies for depression as there could be some underlying reason for the better efficacy of serious game therapies for depression therapies.

Additionally, our literature review, as with most other reviews, has the pitfall of not being able to include all the instances of serious game interventions for the affective disorders we encountered in the reviewed articles for both positive, neutral and negative impacts on the outcome measures. This could be due to the fact that many unsuccessful interventions might not have been published in the scientific literature. Hence, our finding of the positive impact of such interventions in 30 out of 37 studies cannot be considered an absolute conclusion in favor of serious game interventions as affective disorder therapies for ASD and neurotypical populations but should be considered as an indication of the effectiveness of such interventions, and more research will be needed to reach a definitive conclusion in this matter. Furthermore, our search strategies seemed to have been too focused on the Emtree synonyms, which meant that only a few most frequently studied affective disorders were present in our search results and so applications of CSGs for other affective disorder therapies could not be included in this study. On a different note, an interesting aspect of investigating serious game applications for affective disorder therapy is identifying the reasons behind effective interventions. Unfortunately, we could not venture in this direction with the reviewed articles as most of them did not investigate or discuss why their interventions were effective in significant detail. This is an avenue of research that would surely be interesting to explore for serious game applications for affective disorder therapies. Furthermore, 12 out of the 37 selected studies employed some form of RCT while 8 of the rest were pilot studies, which indicates the need for much further exploration of serious game applications for affective disorders in ASD and neurotypical individuals.

**Author Contributions:** The following contributions were made by the authors in the respective aspects of producing this research article. Conceptualization, F.A., J.R.C., F.B. and R.B.; Methodology, F.A.; Software, F.A.; Validation, F.A., J.R.C., F.F., R.B., G.B. and F.B.; formal analysis, F.A., J.R.C., R.B., G.B., F.B. and F.F.; Investigation, F.A.; resources, F.A.; data curation, F.A.; writing—original draft preparation, F.A.; writing—review and editing, J.R.C., R.B., G.B., F.B. and F.F.; visualization,

F.A.; supervision, J.R.C., R.B.; project administration, F.A. All authors have read and agreed to the published version of the manuscript.

**Funding:** This research has been made possible due to the funding received from the Joint Doctoral PhD Program between Queen Mary University of London and University of Genoa.

**Institutional Review Board Statement:** Not applicable.

**Informed Consent Statement:** Not applicable.

**Data Availability Statement:** Not applicable.

**Conflicts of Interest:** The funders had no role in the design of the study; in the collection, analyses, or interpretation of data; in the writing of the manuscript; or in the decision to publish the results.

## Abbreviations

The following abbreviations are used in this manuscript:

| | |
|---|---|
| ASD | Autism spectrum disorder |
| DASS-21 | Depression anxiety and stress scale |
| PANAS | Positive and negative affect scale |
| GSES | General self-efficacy scale |
| ERQ | Emotion regulation questionnaire |
| CGI | Clinical global impressions scale |
| C-GAS | Children's global assessment scale |
| CBCL | Child behavior checklist |
| SPAIC-PV | Social phobia and anxiety inventory for children–parent version |
| SCAS | Spence children's anxiety scale |
| STAXi-CA | State-trait anger expression inventory—child & adolescent |
| MOAS | Modified overt aggression scale |
| DBDR | Disruptive behavior rating disorder scale |
| CGI-S | Clinical global impressions severity |
| PHQ-9 | Patient health questionnaire 9 |
| BAT | Behavioral approach test |
| SCID-I/P | Structured clinical interview for DSM-IV axis-I disorders |
| GAD-7 | Generalized anxiety disorder assessment |
| MDAS | Modified dental anxiety scale |
| DFS | Dental fear survey |
| DSM-IV | Diagnostic and statistical manual of mental disorders |
| STAIC | State-trait anxiety inventory for children |
| GDS | Geriatric depression scale |
| BAI | Beck anxiety inventory |
| BDI | Beck depression inventory |
| ATHQ | Attitudes towards heights questionnaire |
| SRI | Stress response inventory |
| HAM-D | Hamilton depression rating scale |
| BDI | Beck depression inventory |
| ANOVA | Analysis of variance |
| POMS2 | Profile of mood state second edition |
| PDS | Post-traumatic diagnostic scale |
| HADS | Hospital anxiety and depression scale |
| NEQ-32 | Negative effects questionnaire |
| RCT | Randomized controlled trial |
| PTSD | Post-traumatic stress disorder |
| SPQ | Spider phobia questionnaire |
| CSG | Computer-based serious game |

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
