# Peer review of "Applications of Serious Games as Affective Disorder Therapies in Autistic and Neurotypical Individuals: A Literature Review"

_applsci, doi:10.3390/app13084706_

Round 1

Reviewer 1 Report

I would like to thank the authors for their contributions to science. The article is a review of the scientific literature on the use of games in affective disorders. Such comparative studies are needed. They provide a comprehensive base of information about research in a particular area. They shorten the process of finding information and inspire new research. In this case, the researchers show what has been discovered regarding the effectiveness of CSG as a treatment for affective disorders. 

The theoretical introduction was soundly developed. The authors sought answers to 3 research questions. The last one: what are the design features of games that are effective for various affective disorders?, is particularly important for the therapeutic community. 

The research procedure was clearly outlined.  Each of the six stages of the protocol was thoroughly characterized. Tables have been developed in a clear manner. They include references to specific literature.

The analysis of the results provides answers to the questions posed. The authors critically addressed the collected research.

Finally, they pointed out the limitations of the analyses made and pointed out potential directions of the research and areas for further exploration. 

Author Response

Please see attachement.

Reviewer 2 Report

This paper presents the protocol and results of a systematic study on serious games about ASD and Neurotypical Individuals. Although well-written and interesting as a topic, there are some comments to take into account:

1) Section 3.1 (Characteristics of Selected Studies) is somehow difficult to follow.

2) Table Isometric applications sometime are being called 2.5D (like in [52]). Authors use 2D in some cases (e.g. Tetris [46] and Flowy [59]), while others 3D (e.g. BIPOLIFE [43]).

3) "Neuro Racer" game [38][57] is described as "First person" and "immersive environment". How is possible to be 2D? Same for "New Horizon and SpaceControl" [64].

4) Sometimes it is not clear what the health context is about. For example, how "Hit the Cancer" [52] is related to ASD? Is this about a game designed for Autistic people who have cancer?

5) Some minor language issues/typos (e.g. "applicability of of serious games", page 39.

6) Why 3D(VR) games were not included in the 3D category? (Figure 3).

7) Not sure what we see in Figure 4. Where is 2D games represented? Moreover, first person are immersive by default. Please explain.

8) "Therefore, it can be concluded that games created for affective disorder therapies are effective" (page 42). The fact that 30 out of 37 studies have positive health impact cannot be a reliable result simply because nobody knows how many games with no positive impact were designed but never published as articles in the scientific literature. Please rephrase this paragraph.

9) "implementation of 3D, First Person/Immersive and Racing games could be suitable as interventions for depression" (page 42). Please avoid overinterpretation. We do not know the reason behind that 3D selection. This finding has to be explained first by a theory before proposing it a rule. Nobody can exclude randomness, or technological preferences of software developers. The section "Therefore... comorbidities" has to be removed, or rephrased.

Reviewer 3 Report

The article presents a qualitative literature review of serious game applications as therapy interventions for different affective disorders in autistic and neurotypical individuals. 

Critical remarks from the reviewer

  • Authors have to format the paper and position the figures after mentioning them in the text. Figures 1, 2, 3, and 4 appear before their announcement in the main text.
  • The methodology and results are presented in detail, though the paper would benefit from widening the analysis on questions like What makes the games effective in the studied affective disorders?
  • The tables take up too much space. For example, authors can shrink Table 2 by uniting the rows which describe the same game (MindLight, Tetris, etc.). Repeating the same information is not necessary. 
  • One of the main characteristics of the games is the interaction with objects or the game field. Table 2 presents a game that points a game mechanics: “Player is just an observer” (p. 14). What makes this program a game instead of a video?
  • The paper needs minor language corrections.
